# Effects of Disinfection and Steam Sterilization on the Mechanical Properties of 3D SLA- and DLP-Printed Surgical Guides for Orthodontic Implant Placement

**DOI:** 10.3390/polym14102107

**Published:** 2022-05-21

**Authors:** Silvia Izabella Pop, Mircea Dudescu, Sorin Gheorghe Mihali, Mariana Păcurar, Dana Cristina Bratu

**Affiliations:** 1Orthodontic Department, Faculty of Dental Medicine, George Emil Palade University of Medicine, Pharmacy, Science, and Technology of Targu Mures, 38 Gh. Marinescu Str., 540139 Târgu Mureș, Romania; silvia.pop@umfst.ro (S.I.P.); mariana.pacurar@umfst.ro (M.P.); 2Department of Mechanical Engineering, Technical University of Cluj-Napoca, 28 Memorandumului Street, 400114 Cluj-Napoca, Romania; 3Department of Prosthodontics, Faculty of Dentistry, “Vasile Goldis” Western University of Arad, 94 Revolutiei Blvd, 310025 Arad, Romania; 4Department of Orthodontics and Dento-Facial Orthopedics, Orthodontic Research Center, Faculty of Dental Medicine, Victor Babeș University of Medicine and Pharmacy, 2 Eftimie Murgu Square, 300041 Timișoara, Romania; bratu.cristina@umft.ro

**Keywords:** 3D printing, surgical guide, sterilization, temporary anchorage devices, flexural strength

## Abstract

Three-dimensional printed surgical guides increase the precision of orthodontic mini-implant placement. The purpose of this research was to investigate the effects of disinfection and of two types of autoclave sterilization on the mechanical properties of 3D printed surgical guides obtained via the SLA (stereolithography) and DLP (digital light processing) printing methods. A total of 96 standard specimens (48 SLA and 48 DLP) were printed to analyze the tensile and flexural properties of the materials. A total of 80 surgical guide (40 SLA and 40 DLP) specimens from each printing method were classified into four groups: CG (control group); G1, disinfected with 4% Gigasept (Gigasept Instru AF; Schülke & Mayer Gmbh, Norderstedt, Germany); G2, autoclave-sterilized (121 °C); and G3, autoclave-sterilized (134 °C). Significant differences in the maximum compressive load were determined between the groups comprising the DLP-(*p* < 0.001) and the SLA- (*p* < 0.001) printed surgical guides. Groups G2 (*p* = 0.001) and G3 (*p* = 0.029) showed significant parameter modifications compared with the CG. Disinfection with 4% Gigasept (Gigasept Instru AF; Schülke & Mayer Gmbh, Norderstedt, Germany) is suitable both for SLA- and DLP-printed surgical guides. Heat sterilization at both 121 °C and 134 °C modified the mechanical properties of the surgical guides.

## 1. Introduction

Orthodontic mini-implants, which are used for skeletal anchorage, have overcome the disadvantages of a lack of patient compliance and undesirable dental movements due to losses in anchorage during orthodontic therapy [1,2,3,4,5,6,7,8,9,10,11,12,13]. Their stability is strongly related to the precision of insertion and the bone thickness at the area of placement [2,3,6,7,8,9,10,11,12,13,14,15,16,17,18,19,20,21]. Previous studies have demonstrated that the use of surgical guides based on cone beam computed tomography (CBCT) imaging for mini-implants placed in the anterior palate have increased the success rates of temporary anchorage devices (TADs), obtaining precise control of the orthodontic force and decreased treatment time [3,6,7,10,14,15].

Nowadays, several 3D printing technologies are available for the additive manufacturing of surgical guides. The stereolithography (SLA) and digital light processing (DLP) techniques are the most frequently used for manufacturing surgical guides for skeletal anchorage [22,23,24,25,26,27,28,29,30,31,32,33,34]. Both processes work by selectively exposing a liquid resin to a light source. SLA uses a moving UV (ultraviolet) laser beam, while DLP uses stationary UV light from a projector to create highly accurate objects with fine-grained geometries [5,6,7,8,9,10,11,12,13,34].

During the insertion of mini-implants, the surgical guides come in contact with blood and saliva; therefore, proper sterilization or disinfection is needed to avoid infection, which negatively affect the stability of the TADs [2,8,11,12,13]. Sterilization is defined as the complete destruction of all microbial life by physical or chemical processes, while disinfection eliminates microorganisms that form bacterial spores [35]. According to the standard ISO 17664, information regarding recommended disinfection and sterilization methods for the materials used in medical devices should be included in the data sheet for the material provided by the manufacturer [36,37,38]. Chemical disinfection with isopropyl alcohol (70%) and autoclave sterilization are recommended by many 3D printing material suppliers, and it is important to investigate the effects of these methods on the mechanical properties of the materials and their behavior during clinical application [3,14,22,23,24,25,26,27,28,29,30,31,32,33,34,35,36,37,38].

Although several studies have focused on the dimensional accuracy of 3D printed surgical guides for the placement of osseointegrated implants, the effects of sterilization on the mechanical properties of 3D printed guides for orthodontic mini-implants have not yet been studied, to the best of our knowledge [4,6,9,10].

The aim of this study was to investigate the effects of disinfection and autoclave sterilization at different temperatures on the mechanical properties (tensile and compressive) of 3D printed surgical guides obtained via SLA and DLP printing as well as to compare the properties of the two types of guides and to determine which the sterilization/disinfection methods is the best for preserving the clinical performance of the surgical guides obtained in the present work.

## 2. Materials and Methods

The mechanical behavior of a complex geometrical element such as a surgical guide that is subjected to mechanical loading after thermal and chemical treatments cannot be understood if the response of the material is not priorly investigated on a standard specimen.

### 2.1. Standard Specimen Testing

Ninety-six standard specimens were fabricated according to ASTM D638-14 (Standard Test Method for Tensile Properties of Plastics) and ASTM D790-03 (Standard Test Methods for Flexural Properties of Unreinforced and Reinforced Plastics) to analyze the tensile and flexural properties of the materials used for the two different types of printing methods. A total of 48 specimens were printed using the DLP method (Asiga Max UV, Asiga, Sidney, Australia), and 48 specimens were printed using the SLA method (Form 2, Formlabs, Boston, MA, USA) and the recommended materials for surgical guides (DentaGuide, Asiga and Dental SG Resin, Formlab, respectively). The specimens created using each printing method were classified into four groups (six samples in each group): CG (control group); G1, disinfected with 4% Gigasept (Gigasept Instru AF; Schülke & Mayer Gmbh, Norderstedt, Germany) for 60 min; G2, autoclave-sterilized (+1 bar, 121 °C, 20 min); and G3, autoclave-sterilized (+2 bar, 134 °C, 10 min).

Standard procedures for testing flexural and tensile proprieties recommend evaluating at least five specimens for each sample in the case of isotropic materials or molded specimens. The sample size of six specimens per group took into consideration both previous studies [12,29] focused on the mechanical proprieties of similar 3D printed polymers (4 or 5 specimens per group) and the obtained results after testing the minimum number of samples required in standards. The distribution of the values representing the mechanical proprieties (strength, strains and modulus of elasticity) reflected by their standard deviation showed good reproducibility of the mechanical tests and relevance of the results.

The samples were evaluated by a three-point flexural test using an Instron 3366 (Instron, type 3366, Norwood, MA, USA) universal test machine. Each sample was placed on a support that allowed lateral movement and was loaded using a loading nose midway between the supports (Figure 1a). The load was applied at a testing speed of 5 mm/min until the sample broke.

The test enabled the flexural strength (maximum flexural stress sustained by the specimen during the bending test), flexural strain (corresponds to flexural strength), and flexural modulus of elasticity to be determined. All values were calculated according to the ASTM D790-03 standard, with the presented values representing the mean value of the six specimens in each group.

A tensile test was performed using a Type V specimen (Figure 1b) according to the standard ASTM D638-14. During the test, the specimens were loaded at a speed of 1 mm/min. The following tensile properties of the material were measured: tensile strength (tensile stress at maximum load), tensile strain at maximum load, and tensile modulus of elasticity.

The data were statistically processed using SPSSv17 software. To analyze the numerical variables, descriptive statistics were performed. The non-parametric Kruskal–Wallis test was applied to compare more than two numerical series without Gaussian distribution. To compare two independent sets of values, the Mann–Whitney U test was applied. The results were considered significant at *p* < 0.05. A post-hoc G Power test, for Mann–Whitney U family tests, two tails, with a Laplace Parent Distribution, 80% power, 0.05 level of significance, 1 as an allocation ratio, and 1.5 as effect size was performed.

### 2.2. Surgical Guide Testing

In order to create a surgical guide, clinical data records from a patient were required. The archest to provide a virtual cast needed to be scanned intraorally, and a CBCT of the maxilla was required. A digital setup determining the placement of the mini-implants and a virtual design of the surgical guides were made using Easy Driver 2.0 (Uniontech 2020) (Figure 2a,b). The design of the surgical guide was exported as an STL file and served as a template for the printing machine. The SLA method (Form 2, Dental SG Resin, Formlabs, Boston, MA, USA) was used to manufacture 40 surgical guides made of medical class 1 resin (Dental SG Resin, Formlabs) certified for dental use (EN-ISO 10993-1:2009/AC:2010, USP Class VI) by printing 0.05 mm thin layers. The average printing time required for each sample was 70 min. The DLP method (DentaGuide Asiga Max UV, Asiga, Sidney, Australia) was used to print 40 surgical guides at an average printing time of 55 min at a 385 nm LED wavelength.

A post-processing sequence consisting of rinsing the samples in alcohol (99% isopropanol) for 5 min, air-drying them, and light-curing them for 30 min (405 nm) at 60 °C was performed before removing the printing supports.

For each printing method, the surgical guides were divided in four groups: CG, G1, G2, and G3 (10 guides for each group), and these groups underwent the same sterilization and disinfection protocol as the standard specimen groups mentioned above.

The surgical guides were tested by placing them between two flat surfaces (compression platens) and applying a compression force, as seen in Figure 3. The guides were subjected to combined loading (compressive loading was accompanied by bending). Due to their geometry, the guides did not undergo rigid body movement during testing, nor was there a change in the points that were in contact with the platens. Force was continuously applied at a speed of 1 mm/min until the guide broke. The maximum compressive load (N) and compressive extension at the maximum compressive load (mm) were measured. Testing a specimen without a constant cross-sectional area results in invalid strain and stress values. In such cases, a comparison of the load–displacement curves is relevant to characterize the mechanical behavior of the parts as long as the material was previously investigated on standard specimens.

The data were statistically processed using SPSSv17 software. To analyze the numerical variables, descriptive statistics were performed. The non-parametric Kruskal–Wallis test was applied to compare more than two numerical series without Gaussian distribution. To compare two independent sets of values, the Mann–Whitney U test was applied. The results were considered significant at *p* < 0.05.

## 3. Results

A total number of 96 specimens and 80 surgical guides were tested.

### 3.1. Standard Specimen Testing

#### 3.1.1. Flexural Test

The values of the flexural strength, flexural strain, and flexural modulus of elasticity parameters of the four groups (CG–G3) of the DLP- and SLA-printed specimens are presented in Table 1.

Regarding the DLP printing method, the comparison of the in-group values showed statistically significant increases in the flexural strength (*p* = 0.007) and flexural modulus (*p* = 0.002) values of group G2 compared to those of the CG. When compared to group G3, both the flexural strength (*p* = 0.041) and the flexural strain (*p* = 0.002) values of group G2 were significantly increased, while the flexural modulus (*p* = 0.002) value decreased. The values of the measured parameters in group G1 did not differ significantly (*p* = 0.335) from those of the control group (CG) (Figure 4).

The descriptive statistics comparing the numerical data of the DLP- and SLA-printed specimens and the *p* value after applying the Mann–Whitney U test are shown in Table 2.

#### 3.1.2. Tensile Test

The tensile strength, tensile strain, and tensile modulus of elasticity values for the four groups (CG–CG3) of DLP- and SLA-printed specimens are presented in Table 3.

The values for the tensile modulus of elasticity (*p* = 0.002) and the tensile strength (*p* = 0.002) for group G2 (DLP method) increased significantly compared to those of the CG (DLP method), and the tensile strain increased, but not statistically significantly (*p* = 0.240). The same modifications were determined for group G3 (DLP method) compared to the CG (DLP method) (tensile strength and tensile modulus of elasticity *p* = 0.002). The statistical analysis did not reveal significant differences in the measured parameters between G1 (DLP method) and the CG (DLP method). When compared to the specimens that were autoclaved at 121 °C (G2), the group of specimens that had been autoclaved at 134 °C (G3) showed significant increases in the tensile strength (*p* = 0.041) tensile modulus of elasticity (*p* = 0.002) values and decreased tensile strain values (*p* = 0.041).

Regarding the SLA printing method, the autoclaved specimens (G2 and G3) showed a significant decrease in tensile strain (*p* = 0.009) and an increase in tensile strength (*p* = 0.041) and tensile modulus of elasticity (0.002) compared to the control group (CG). The values of the same parameters for G2 and G3 showed no significant differences (Figure 5).

The descriptive statistics comparing the numerical data of the DLP- and SLA-printed specimens and the *p* value after applying the Mann–Whitney U test are shown in Table 4.

### 3.2. Surgical Guide Test

A comparison of the load–displacement curves for the surgical guides fabricated using the DLP and SLA methods is presented in Figure 6. The obtained curves display a brittle attitude since the majority of these elements had a quasi-linear behavior until break with reduced deformation.

The values of the maximum compressive load and compressive extension at the maximum compressive load are shown in Table 5.

Significant differences in the maximum compressive load were found between groups for both the DLP- (*p* < 0.001) and SLA- (*p* < 0.001) printed surgical guides. This parameter was significantly increased when the DLP-printed surgical guides underwent autoclave sterilization (groups G2, *p* = 0.001 and G3, *p* = 0.029). For the DLP-printed guides, the decrease in the compressive extension at the maximum compressive load was relevant in group G3 when compared to the control group (CG) (*p* < 0.001).

The maximum compressive load of the SLA surgical guides showed a significant decrease after autoclave sterilization (G2, *p* < 0.001 and G3, *p* = 0.003). The compressive extension at the maximum compressive load of the autoclave-sterilized guides (G2 and G3, *p* < 0.001) revealed the same modifications.

When compared to the control group (CG) of the DLP-printed surgical guides, the control group (CG) of the SLA-printed surgical guides showed significantly increased (*p* = 0.01) compressive load (Figure 7a) and compressive extension at the maximum compressive load (*p* = 0.012) values (Figure 7b).

## 4. Discussion

This study aimed to determine the mechanical properties of surgical guides that are commonly used for mini-implant placement after undergoing chemical disinfection and autoclave sterilization. The characterization of different materials from the point of view of their mechanical properties is extremely important to understand their behavior in a clinical situation and to understand how the different procedures that are applied before their clinical use (such as disinfection and sterilization) might affect their clinical performance [4,12,13,39,40,41,42,43,44,45].

Printed surgical guides are becoming more and more common in everyday orthodontic practice to increase the precision of mini-implant placement [3,38,39,46,47,48,49,50,51]. However, sterilization or disinfection of these medical devices is highly recommended and regulated [2,8,11,12,13,51], and so far, the effects of chemical disinfectants and heat sterilization on the mechanical properties of surgical guides have not been investigated in depth. A few studies in the literature [11,12,13,42] have investigated the effects of sterilization and disinfection on drilling templates. A study by Smith et al. recommended disinfecting surgical guides with 70% ethanol for 15 min [42], but the mechanical behavior of the disinfected guides was not evaluated. In our study, the specimens were disinfected (G1) using 4% Gigasept (Gigasept Instru AF; Schülke & Mayer Gmbh, Norderstedt, Germany) for 60 min. The results of the flexural and tensile tests of both disinfected specimens that were SLA- and DLP-printed, showed no statistical differences in the studied parameters compared to the guides in the CG. When the surgical guides were evaluated, the SLA-printed guides in G1 showed an increased maximum compressive load compared to the CG, while the compressive extension at the maximum compressive load was not affected. Torok et al. [8] conducted flexure and compressive tests on 3D printed surgical templates made of Objet MED 610 (Stratasys). They concluded that disinfection did not modify the evaluated parameters.

When comparing the SLA-printed guides with the DLP-printed ones, the maximum compressive load and compressive extension at maximum of the compressive load were significantly increased in the CG. The differences were also noticeable when the standard specimens were compared. This can be explained by the differences in the material and printing method and can be clinically translated as the SLA-printed guides having higher strength [8,9,10]. The basic tensile strength and elastic modulus of printed components produced with SLA printers were investigated in several studies [27,40,41]. The results showed some distinctions between the tensile modulus of 3D prints and their base materials. The tensile properties of specimens with edge build orientation are different compared with those of specimens with flat build orientation [41,52,53].

However, autoclave sterilization (both at 121 °C and 134 °C) decreased the maximum compressive load of the SLA-printed guides. On the other hand, the DLP-printed guides revealed an increase in the maximum compressive load after sterilization. The different geometrical morphology of the guides when compared to the standard specimens and the differences between the printing methods might explain the decrease in the maximum compressive load of the SLA-printed guides. The specimens used for the tensile and flexural tests were fabricated according to specific required standards (ASTM D638-14 and ASTM D790-03). On the other hand, surgical guides have a different geometrical morphology when compared to a standard specimen. The standard specimens are flat, while the surgical guides have a convex morphology required by the shape of the maxillary palatum. Quintana [52] stated that the mechanical properties of photo-curable resins used for SLA printing are influenced by the build orientations (flat or edge) of the tested parts. Kazemi [53] observed that the supporting structures influence the tensile strength of stereolithography (SLA)-fabricated parts by increasing the roughness of the sample surface.

In contrast to our study, Torok et al. stated that steam sterilization at 121 °C and plasma sterilization had no significant effects on the dimensional changes and properties of the material of the tested drill templates [8]. Shaheen et al. observed noticeable morphological changes in the orthognathic splints after sterilization, with heat sterilization making the splints less reliable [12].

The analysis of the measurement results for the standard specimens underlined the thermal effects that take place during the sterilization procedure on the material’s behavior. Both materials became more brittle when tensile and bending stresses increased as the corresponding strains increased. This effect was also visible when examining the evolution of the elastic moduli (tensile and flexural), which increased with respect to those of the specimens that were not subjected to thermal treatment. Torok et al. [8] stated that autoclaving at 134 °C increased the compressive force, but disinfection decreased it by 36%.

It was not only the temperature value that influenced the mechanical behavior, but also the exposure time of the materials at that temperature. Both DLP- and SLA-printed materials became more sensitive from the point of view of the elastic modulus after sterilization, with a longer time producing t significant increase in the modulus, resulting in a stiffer material. How the material behaved due to thermal treatment as observed during the tensile and bending tests was also reflected in the maximum amount of force supported by the surgical guides and their corresponding displacements. The load–deflection curves showed a clear brittle behavior of the surgical guides after autoclave sterilization, especially for the guides made by the SLA technology, with a significant decrease in the displacements. Thermal sterilization produced an increase in stiffness of all guides, a higher sterilization temperature (group G3) leading to a stiffer guide.

The main limitations of our study regard the in vitro design. An evaluation of the clinical behavior of the surgical splints might be useful for obtaining more accurate data. Increasing the sample size and expanding the sterilization methods to include plasma sterilization would also provide important data for further studies. Evaluation of the surface properties and dimensional accuracy of the specimens is important in assessing the effects of sterilization and disinfection, and further studies are required to formulate final conclusions.

It can be stated that according to our results, autoclave sterilization significantly modifies surgical guides when these are subjected to combined loading (compressive accompanied by bending). Clinically, this modification might be relevant when the surgical guide is firmly applied and pressed towards the dental arches in terms of the increased rigidity of the material and possible fracture during use.

## 5. Conclusions

Whitin the limitations of this study, the following conclusions can be drawn:Disinfection with 4% Gigasept (Gigasept Instru AF; Schülke & Mayer Gmbh, Norderstedt, Germany) is suitable both for SLA- and DLP-printed surgical guides.Heat sterilization at both 121 °C and 134 °C modifies the mechanical properties of the surgical guides and is not recommended as a sterilization method for surgical guides intended for mini-implant placement.

## Figures and Tables

**Figure 1 polymers-14-02107-f001:**
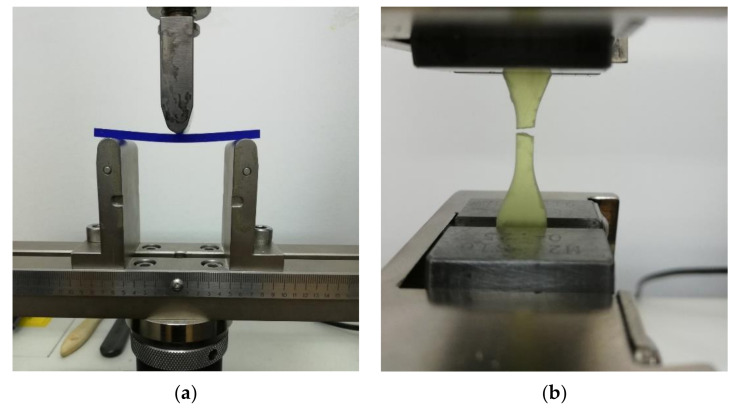
Specimens being tested to determine the mechanical characteristics of the investigated materials: (**a**) bending the specimen in a three-point bending test; (**b**) tensile specimen (Type V) after the tensile test.

**Figure 2 polymers-14-02107-f002:**
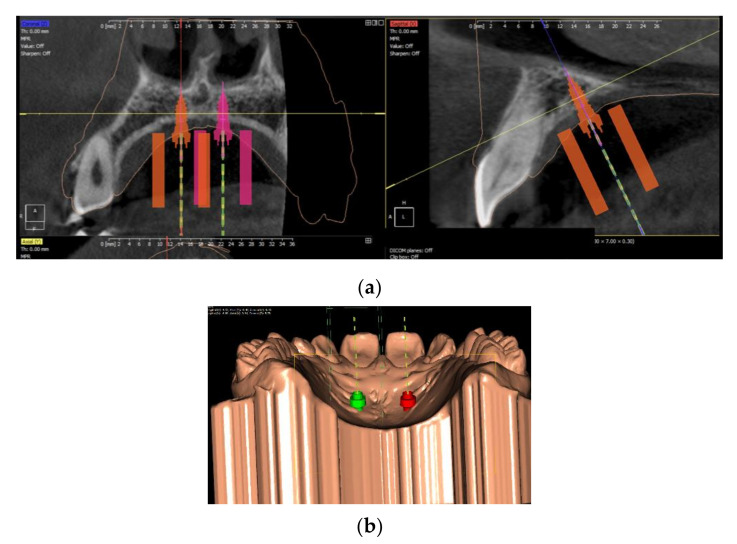
Digital setup determining the placement of the mini-implants and the virtual design of the surgical guides: (**a**) virtual placement of the mini-implants on the CBCT; (**b**) positioning of the mini-implants on the virtual cast.

**Figure 3 polymers-14-02107-f003:**
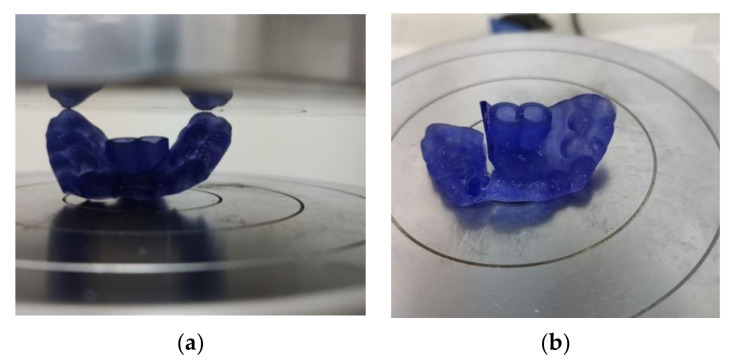
Experimental investigation of the surgical guides: (**a**) compression test on a guide placed between compression platens; (**b**) failure of the surgical guide subjected to compression force.

**Figure 4 polymers-14-02107-f004:**
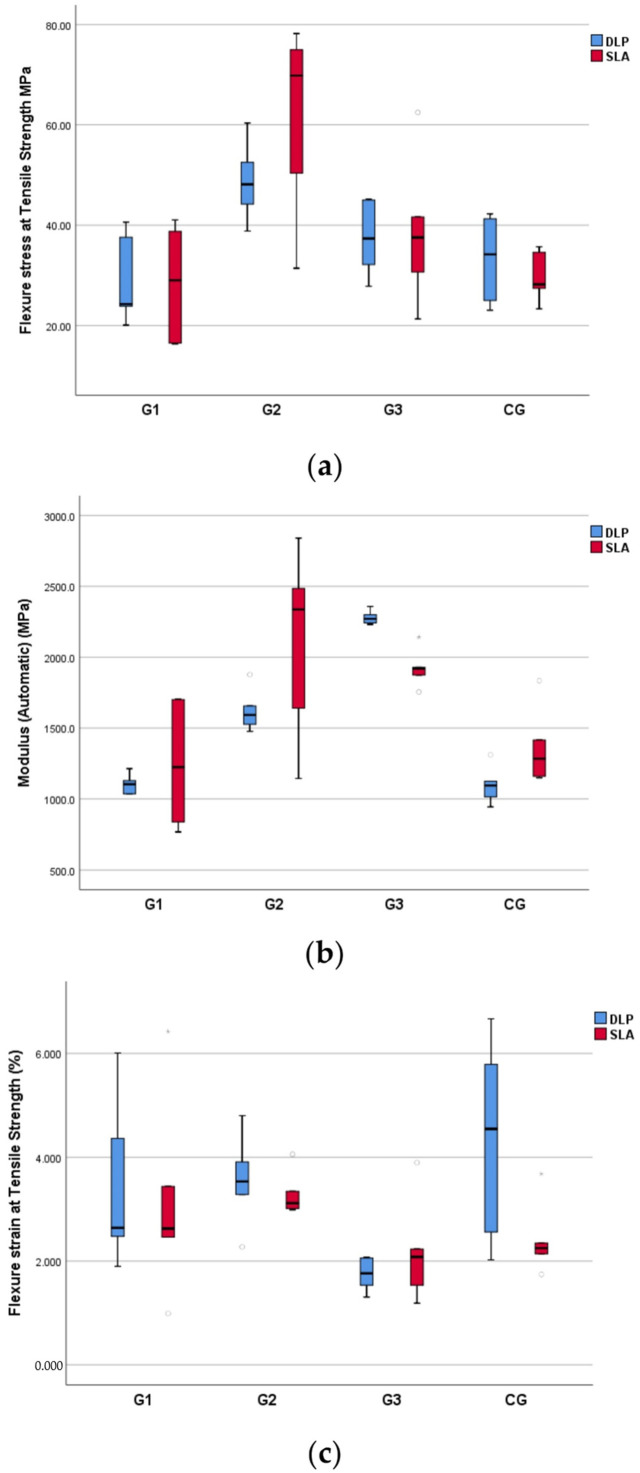
Boxplots of minimum, maximum, interquartile range, median, and outliers for standard specimens (CG–G3) during the flexural test: flexural modulus of elasticity (**a**); flexural strength (**b**); flexural strain (**c**).

**Figure 5 polymers-14-02107-f005:**
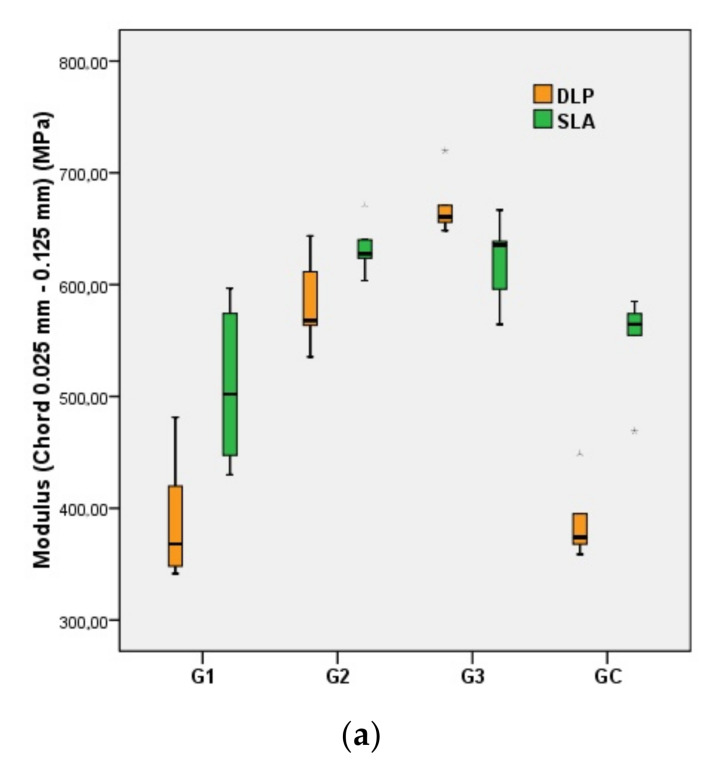
Boxplots of minimum, maximum, interquartile range, median, and outliers for standard specimens (CG–G3) in the tensile test: tensile modulus of elasticity (**a**); tensile strength (**b**); tensile strain (**c**).

**Figure 6 polymers-14-02107-f006:**
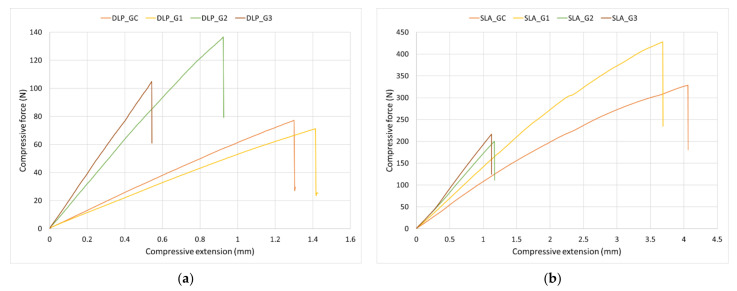
Compressive load–displacement curves: (**a**) surgical guides printed using the DLP method; (**b**) surgical guides printed using the SLA method.

**Figure 7 polymers-14-02107-f007:**
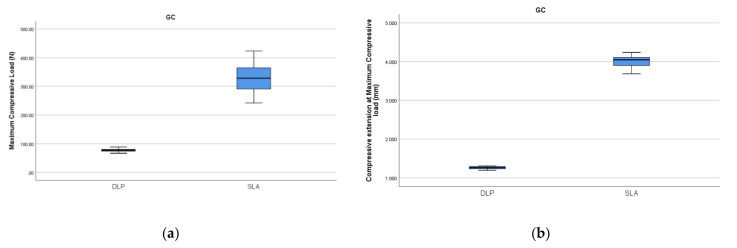
Boxplots of the comparisons between the CG: compressive load (**a**) and compressive extension at the maximum compressive load (**b**).

**Table 1 polymers-14-02107-t001:** Mean and standard deviations of the measured parameters of the specimens obtained during the flexural test.

Groups	Printing Method	Flexural Strength (MPa) Mean ± Std Dev	Flexural Strain % Mean ± Std Dev	Flexural Modulus of Elasticity (Mpa) Mean ± Std Dev
CG	DLP	33.33 ± 8.21	4.36 ± 1.80	1097.35 ± 124.23
	SLA	29.58 ± 4.71	2.40 ± 0.66	1354.57 ± 257.72
G1	DLP	28.46 ± 8.46	3.34 ± 1.55	1103.55 ± 67.63
	SLA	28.45 ± 11.18	3.09 ± 1.81	1244.07 ± 432.33
G2	DLP	48.69 ± 7.36	3.56 ± 0.83	1620.13 ± 140.14
	SLA	62.44 ± 18.02	3.27 ± 0.41	2130.73 ± 623.57
G3	DLP	37.48 ± 7.07	1.75 ± 0.30	2278.78 ± 47.06
	SLA	38.53 ± 13.87	2.17 ± 0.94	1923.30 ± 125.30

**Table 2 polymers-14-02107-t002:** Flexural strength, flexural strain, and flexural modulus of elasticity values of the DLP- and SLA-printed specimens.

Specimen Type	Flexural Strength (MPa) Mean ± Std Dev	Flexural Strain % Mean ± Std Dev	Flexural Modulus of Elasticity (MPa) Mean ± Std Dev
DLP	36.99 ± 10.54	3.25 ± 1.53	1524.95 ± 503.72
SLA	39.75 ± 18.42	2.73 ± 1.12	1663.17 ± 536.99
*p* value	0.821 ^is^	0.375 ^is^	0.257 ^is^

^is^—insignificant difference.

**Table 3 polymers-14-02107-t003:** Mean and standard deviations of the measured parameters for the specimens during the tensile test.

Groups	Printing Method	Tensile Strength (Mpa) Mean ± Std Dev	Tensile Strain % Mean ± Std Dev	Tensile Modulus of Elasticity (MPa) Mean ± Std Dev
CG	DLP	30.28 ± 3.36	13.61 ± 2.03	386.48 ± 32.96
	SLA	41.17 ± 8.55	12.35 ± 3.17	551.98 ± 42.30
G1	DLP	29.66 ± 4.30	11.96 ± 1.55	387.81 ± 54.23
	SLA	36.38 ± 8.88	10.54 ± 2.87	508.74 ± 72.64
G2	DLP	45.56 ± 3.89	12.02 ± 1.18	581.63 ± 38.94
	SLA	29.61 ± 9.65	5.58 ± 2.39	632.30 ± 22.38
G3	DLP	51.42 ± 2.57	10.78 ± 0.57	669.42 ± 25.90
	SLA	23.74 ± 5.48	4.18 ± 1.22	622.79 ± 36.43

**Table 4 polymers-14-02107-t004:** Tensile strength (tensile stress at maximum load), tensile strain at maximum load, and tensile modulus of elasticity values of the DLP and SLA printed specimens.

Specimen Type	Tensile Strength (MPa) Mean ± Std Dev	Tensile Strain % Mean ± Std Dev	Tensile Modulus of Elasticity (MPa) Mean ± Std Dev
DLP	39.23 ± 10.26	12.09 ± 1.69	506.34 ± 131.07
SLA	32.72 ± 10.27	8.16 ± 4.18	578.95 ± 68.21
*p* value	0.028 ^s^	0.005 ^s^	0.087 ^is^

^is^—insignificant difference; ^s^—significant difference.

**Table 5 polymers-14-02107-t005:** Mean and standard deviations of the maximum compressive load and compressive extension at the maximum compressive load for the tested surgical guides.

Groups	Printing Method	Maximum Compressive Load (N) Mean ± Std Dev	Compressive Extension at Maximum Compressive Load (mm) Mean ± Std Dev
CG	DLP	77.30 ± 7.08	1.26 ± 0.04
	SLA	329.41 ± 62.39	4.00 ± 0.19
G1	DLP	74.45 ± 6.91	1.33 ± 0.11
	SLA	413.66 ± 43.97	3.98 ± 0.33
G2	DLP	145.22 ± 34.67	1.06 ± 0.28
	SLA	165.86 ± 43.11	0.95 ± 0.26
G3	DLP	122.92 ± 37.75	0.55 ± 0.08
	SLA	225.24 ± 26.88	1.14 ± 0.13

## Data Availability

Not applicable.

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
