# Peer review of "Effects of Disinfection and Steam Sterilization on the Mechanical Properties of 3D SLA- and DLP-Printed Surgical Guides for Orthodontic Implant Placement"

_polymers, 2022, doi:10.3390/polym14102107_

Round 1
Reviewer 1 Report
01
There were 6 samples in each group, and yet the authors used ANOVA to compare the mean values between the groups. If the number of samples per group is low, the analysis calls for a non-parametric test regardless of the normality, as the normal distribution cannot properly be verified. Moreover, even if normality can be reasonably assumed, in small samples tests that assume normally distributed data are likely to be underpowered to detect departures from the equal variance assumption. That is, use of these tests in small samples may lead researchers to incorrectly conclude that the equal variance assumption is justified. Therefore, the authors will have to redo the entire statistics, but now applying a non-parametric test instead of ANOVA.
02
There are some sentences in the text without reference to a previous study (or studies) in order to give evidence to their statements. Without references, these statements would be mere assumptions or allegations by the author of the thesis. Therefore, each of the following sentences need at least one reference to back up their statement:
“The increase in the maximum compressive load of the SLA-printed guides in G1 might be explained by a chemical reaction that takes place between the extended surface of the surgical guide and the solution used for disinfection.”
“The different geometrical morphology of the guides when compared to the standard specimens and the differences between the printing methods might explain the decrease in the maximum compressive load of the SLA-printed guides.”
03
“The increase in the maximum compressive load of the SLA-printed guides in G1 might be explained by a chemical reaction that takes place between the extended surface of the surgical guide and the solution used for disinfection.”
Which chemical reaction? How can this chemical reaction possible affect the maximum compressive load?
“This can be explained by the differences in the material and printing method and can be clinically translated as the SLA-printed guides having higher strength.”
Which differences in the material? Which differences in the printing method? The authors have not discussed that.
04
No power analysis was performed. Therefore, it is not possible to know whether the analysis of the results of the present study is a true finding or a pure chance. This may compromise the entire validity of this study.
05
The following paragraph is a repetition of the Results without an actual discussion of the findings:
“The analysis of the measurement results for the standard specimens underlined the thermal effects that take place during the sterilization procedure on how the material behaves. Both of the materials became more brittle when tensile and bending stresses increased as the corresponding strains increased. This effect was also visible when examining the evolution of the elastic moduli (tensile and flexural), which increased with respect to the specimens that were not subjected to thermal treatment. It was not only the temperature value that influenced the mechanical behaviour, but also the exposure time of the materials at that value. Both of the DLP- and SLA-printed materials became more sensitive from the point of view of the elastic moduli after sterilization, with a longer time producing ta significant increase in the modulus, resulting in a stiffer material. How the material behaved due to thermal treatment as observed during the tensile and bending tests is also reflected in the maximum amount of force supported by surgical guides and their corresponding displacements.”
06
The limitations of the study were neither pointed out nor discussed.
Reviewer 2 Report
The presented study entitle "Effects of Disinfection and Steam Sterilization on the Mechanical Properties of 3D SLA- and DLP-Printed Surgical Guides for Orthodontic Implant Placement" was focused on the evaluation SLA/DLP printed parts properties after sterilization.
The subject of research cannot be consider as novel, however, the proposed studies may be helpful as a comprehensive comparative material for further research in this area. After major correction the manuscript can be processed as journal paper. My comments are highlighted below.
Mechanical tests indicate very large differences in the behavior of individual samples, for example flexural strength for DLP-CG sample was 20.24±18.96.... is there any clear reason for these deviations or how to avoid this lack of repetition.
Some of the mechanical measurements include compression tests of finished products, it would be advisable to discuss the type of damage occurring, and present the nature of the crack (brittle or ductile).
Tests should present details of the appearance of the samples, before and after sterilization. In current version, the research work is mostly focused on the results of mechanical measurements, while the surface quality may also be crucial in assessing the effects of sterilization.
Authors should supplement the presented research with an analysis of the accuracy of the products shape, so that changes in geometry and dimensions after the sterilization process can be visualized.
Round 2
Reviewer 1 Report
“We made a G Power test, for Mann-Whitney U family tests, two tails, with a Laplace Parent Distribution, 80% power, 0.05 level of significance and 1 as an allocation ratio.”
80% power, alpha 5%, and allocation ratio of 1 are standard values in a power analysis. And it is impossible to calculate the sample size without an effect size, which the authors have not provided. Therefore, the authors provide me this answer without actually doing anything.
Reviewer 2 Report
The introduced corrections can be considered sufficient, the article may be published in its current form. However, I suggest that in the case of subsequent publications, the topic of the influence of sterilization treatment on deformation of products should be developed in more detail.
Author Response
Thank you for your valuable suggestions and comments!
Round 3
Reviewer 1 Report
01
“Similar studies, focused on the mechanical properties of materials, included 3 to 5 specimens for tensile and flexural tests:”
The recklessness from a statistical point of view committed by others do not justify yours.
02
“Five samples have been considered for each specimen type for a total of 25 tested specimens, following the recommendations of ASTM D638-10 Standard”
Please provide a screenshot of the passage of the text in the ASTM D638-10 Standard where it is stated that five samples are enough, and that this would better qualify the analyses of the results as true findings rather than pure chance, from the statistical point of view.
03
“As mentioned before, a post-hoc G Power test, for Mann-Whitney U family tests, two tails, with a Laplace Parent Distribution, 80% power, 0.05 level of significance, 1 as an allocation ratio and 1.5 as effect size was done.”
“Therefore, taking into consideration the abovementioned facts, we considered testing 6 specimens/group.”
An “effect size of 1.5” is new information, added only in the latest version of the manuscript. Please provide literature reference (at least one previous study that analyzed such materials, with enough statistical power, in which the comparison of the groups resulted in an effect size of 1.5) that would justify the choice for an effect size of 1.5. Otherwise, it is just an assumption. Or better saying, a choice of value in order to better fit the power analysis recommendation that would fit the study. And yet, a wrong choice, as a proper power analysis using these parameters would result in 9 samples/specimens per group.
Still, nothing was done, and the authors are trying to lure the reviewer.
04
The authors used which software to perform the power analysis?
05
“The study described in the manuscript was done exactly as described (during 9 month of hard work) and
we accurately presented our research findings. We fully understand the publication ethic statement about
data or image manipulation.”
Yes, I believe that the methodology was accurately described and the study was well-conducted, but the small number of samples does not allow the authors to distinguish the analysis of the results as true findings from pure chance.
